# Machine learning detects hidden treatment response patterns only in the presence of comprehensive clinical phenotyping

Stephen D. Auger[1,2,3]*, Gregory Scott[1,2,3]

1 Department of Brain Sciences, Imperial College London, London, United Kingdom, 2 UK DRI Care Research and Technology Centre, Imperial College London, London, United Kingdom, 3 Department of Neurology, Imperial College NHS Trust, London, United Kingdom

* s.auger21@imperial.ac.uk

## Abstract

Inferential statistics traditionally used in clinical trials can miss relationships between clinical phenotypes and treatment responses. We simulated a randomised clinical trial to explore how gradient boosting (XGBoost) machine learning compares with traditional analysis when 'ground truth' treatment responsiveness depends on the interaction of multiple phenotypic variables. As expected, traditional analysis detected a significant treatment benefit (outcome measure change from baseline = 4.23; 95% CI 3.64–4.82). However, recommending treatment based upon this evidence would lead to 56.3% of patients failing to respond. In contrast, machine learning correctly predicted treatment response in 97.8% (95% CI 96.6–99.1) of patients, with model interrogation showing the critical phenotypic variables and the values determining treatment response had been identified. Importantly, when a single variable was omitted, accuracy dropped to 69.4% (95% CI 65.3–73.4). This proof of principle underscores the significant potential of machine learning to maximise the insights derived from clinical research studies. However, the effectiveness of machine learning in this context is highly dependent on the comprehensive capture of phenotypic data.

## Introduction

The clinical phenotype refers to the observable characteristics of a disease, and is a crucial indicator of both its presence and how it manifests in an individual. A clinical phenotype includes numerous variables, such as a patient's age and other demographic factors, co-morbidities, symptoms, and examination findings. Clinical phenotypes can be expanded by measuring and understanding more variables, including genetic, physical, environmental, radiological, electrophysiological, biochemical, and molecular characteristics, i.e., so-called 'deep phenotyping'. An important motivation for clinical phenotyping is the idea that, somewhere within the phenotype, are variables that inform how a disease can be optimally treated within an individual.

**Data availability statement:** All the datasets generated and analysed during the current study are available in the GitHub repository, https://github.com/stepdaug/ML-and-clinical-phenotyping.

**Funding:** SDA is funded by a UK National Institute for Health and Care Research (NIHR) Clinical Lectureship and acknowledges infrastructure support for this research from the NIHR Imperial Biomedical Research Centre (BRC). GS is funded by the National Institute for Health Research [Advanced Fellowship]. The views expressed are those of the authors and not necessarily those of the NIHR or the Department of Health and Social Care.

**Competing interests:** The authors have declared that no competing interests exist.

For example, the management of headache depends upon numerous, interacting, variables within the clinical phenotype. These variables include whether the headache is migrainous, the presence and type of aura, and the presence of co-morbidities, like gastritis, asthma, or anxiety [1]. Within broad diagnostic categories, there are often distinct clinical phenotypes, which can be driven by different underlying disease processes that require tailored management approaches. Consequently, the optimal treatment strategy for a young female in full-time work with menstrual migraine and aura differs significantly from that of a retired older male with migrainous cervicogenic headache. Even single variables within an individual's phenotype can change an otherwise effective treatment into something likely to cause harm, e.g., carbamazepine as a treatment of epilepsy in people of Han Chinese ethnicity [2].

For new treatments, particularly of heterogenous diseases, it can take decades to determine whether patients with a given clinical phenotype may benefit, or, in fact, not at all [3]. Without this understanding, groups of patients may receive treatments that provide no benefit and/or may cause harm. For example, aspirin was used for decades to prevent cardiovascular disease before evidence showed its benefit in primary prevention was minimal and that additional clinical variables needed to be considered to balance the benefits for high-risk patients against the associated bleeding risks [4,5]. Many similar examples exist, where targeting treatments based on specific clinical phenotypes has become necessary, such as beta-blockers in heart failure [6,7], calcium channel blockers in hypertension [8,9], and antidepressants in major depressive disorder [10]. In each case, certain patient groups have historically unknowingly received ineffective treatments for many years. Understanding and managing the heterogeneity of clinical phenotypes is therefore critical for personalised and effective care. However, building the evidence base to support this level of precision medicine requires time and, crucially, appropriate methodologies that can deconstruct this phenotypic heterogeneity.

Randomised controlled trials (RCTs) are among the highest levels of evidence in clinical research. Designing, funding, conducting, and publishing RCTs takes many years or even decades [11]. Given this investment, it is crucial to maximise the information gained from RCTs, to realise the greatest benefits for patients. Advances in machine learning (ML) and other multivariate analysis techniques offer opportunities to identify complex relationships between clinical phenotype and treatment response, which the inferential statistical methods traditionally used for RCT analysis may miss [12,13]. Indeed, a growing amount of clinical research includes the use of ML techniques, and considers their current limitations when applied in clinical settings [14,15]. The effectiveness of ML models, as for all methodologies, critically depends on the quality and quantity of the data available to them [16].

In this study, we investigated how both the analysis approach and characterisation of clinical phenotype within an RCT influences our ability to infer underlying 'ground truth'. Using simulated data representative of typical RCTs, we conducted a novel, formal comparison contrasting the effectiveness of traditional statistical

methods versus ML approaches at identifying the key factors and interactions driving treatment response. We simulated a clinical cohort undergoing a RCT, providing ground truth information about how variables within the clinical phenotype determine responses to the treatment. We compared this ground truth information with the conclusions that would be drawn by investigators when analysing the RCT data using traditional inferential statistics. We then examined the capability of ML to uncover additional insights from the same data. We used XGBoost (XGB), a form of gradient boosting ML which has demonstrated state-of-the-art performance on a range of problems involving complex, non-linear interactions between variables [17,18], and has been applied in a variety of clinical settings [19–22]. We evaluated the additional benefits of XGB analysis, including the ability to reveal the phenotypic variables and values which critically determine treatment response. Finally, we examined how the comprehensiveness of clinical phenotyping might impact conclusions when using XGB analysis, considering the effects both of data deficiency and excess.

## Results

### Creation of simulated clinical cohort data and determinants of 'ground truth' treatment response

We created simulated clinical cohort data, modelling a group of 1000 patients with a disease for which a new treatment improves outcomes in those with certain clinical phenotypes, but not others. These patients are subsequently enrolled in a parallel group randomised placebo-controlled trial (1:1 randomisation). (See Methods for full details of the cohort's characteristics and modelling of treatment response).

For each patient, the clinical phenotype consists of multiple arbitrary clinical variables, including illustrative variables for patient age, sex, and several additional clinical variables. Both binary and continuous variables are simulated; binary variables being analogous to, e.g., sex or presence/absence of a symptom, and continuous variables analogous to, e.g., age or a numerical symptom severity score.

There are three such clinical variables that critically determine which patients are responsive to the treatment: 'X', 'Y' and 'Z'. X and Y are continuous; Z is binary. This scenario is analogous to, e.g., real-world clinical variables such as (respectively) bone mineral density, age, and sex, which jointly help determine which bone protection interventions are suitable for a given clinical phenotype [23]. Importantly, the investigators in our scenario have no knowledge of this ground truth information about which variables determine treatment response.

A simulated non-linear relationship between these three variables (X, Y, Z) determines which patients will be responsive to the treatment, illustrated in Fig 1 (orange indicates treatment responsive conditions, blue indicates not responsive), and described as follows: Any patient with a value of X above 95 is responsive to the treatment, no matter whether Z is present (left plot) or absent (right plot). For lower values of X (X<=95), if Z is present (left plot) and Y is between 50 and 90 then the patient is treatment responsive. If Z is absent (right plot) and X is between 90 and 95, then a patient is treatment responsive if Y is between 50 and 90.

With these conditions, 43.7% of patients are potentially responsive to treatment; 56.3% are not. Additional information is captured in further variables, but importantly, these have no bearing on treatment response (Fig 1). To aid interpretation, we denote these uninformative variables with numbers (i.e., 'V1' and 'V2') rather than the letters which denote the treatment response-determining variables above.

Treatment response, or non-response, is reflected in a distinct, continuous, outcome measure with arbitrary units. Patients responsive to the treatment will, on average, have a positive change in the outcome measure after receiving treatment; patients not responsive to treatment will, on average, have no change in the outcome measure with treatment. Non-responsive patients include those receiving placebo and those who receive treatment yet who are not responsive to it. To reflect the clinical reality of trial outcome measures having a degree of variability, the change in the trial outcome measure is modelled as being drawn from one of two normal distributions, of mean +10 or mean 0, for treatment responsive and non-responsive patients, respectively. Both distributions have standard deviation 3.

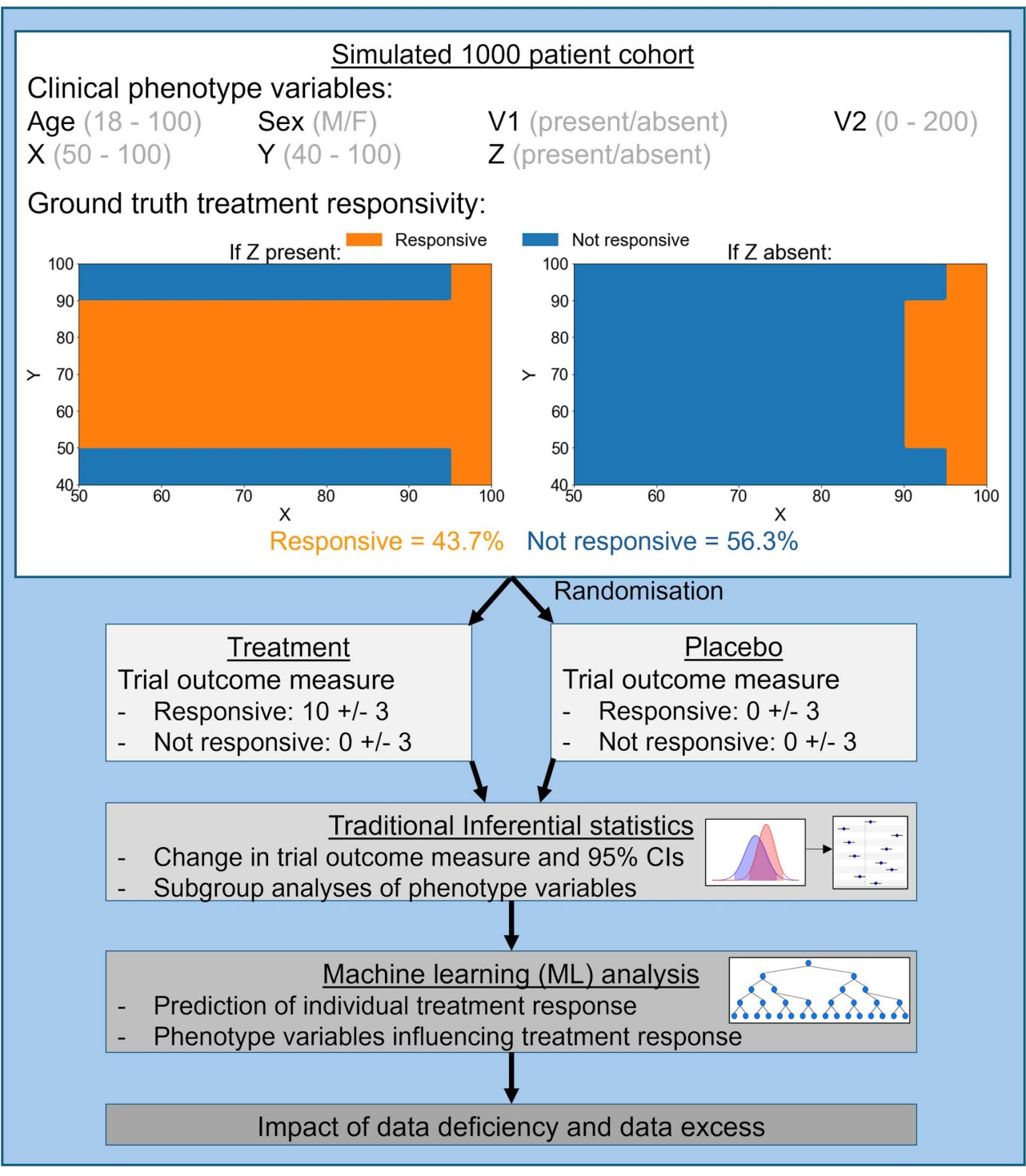

**Fig 1. Investigation outline.** Clinical data were generated for a simulated cohort of 1000 patients. The top panel shows the seven clinical pheno-type variables. The range of values for each variable is shown in brackets. The plots indicate the combination of values of the critical variables determining treatment responsiveness: orange areas indicate values associated with patients being responsive and blue areas indicate non-response to treatment. The left and right plots indicate X and Y values when Z is present or absent respectively. By this arrangement 43.7% are responsive and 56.3% not responsive to treatment. Patients were randomly assigned to a treatment or placebo group (1:1 randomisation), with trial outcomes based on their true responsiveness and assigned group. The numbers in the 'Treatment' and 'Placebo' boxes represent the change in outcome measures

(mean+/- standard deviations) for each group, according to their true responsiveness. We performed traditional inferential statistical analysis on the trial outcome data, to obtain estimates of effect size (mean change) and precision (95% confidence intervals). Then, machine learning (ML) analysis with XGBoost was conducted to predict individual patient treatment responses and to identify which clinical phenotype variables influenced these predictions. Finally, we assessed how data deficiencies and excesses impact ML analysis. CI = confidence interval.

We simulated an RCT in which the 1000 patients are randomised to receive either the treatment or a placebo in a 1:1 ratio. The value for each patient's change in the trial outcome measure is determined by their ground truth treatment responsivity (above) and their treatment/placebo group allocation. Patients of all clinical phenotypes were eligible for inclusion in the study. The RCT was rigorously conducted and well-powered, without loss to follow-up. The investigators collected and have available to them the complete information about all the variables described above (Fig 1), but not the ground truth treatment responsiveness – this information is what the trail aims to infer.

### Analysis using traditional inferential statistics

The RCT was designed to investigate whether a new treatment improves a clinical outcome measure versus placebo. There is clinical cohort data for 1000 patients with the phenotypes described above. Following randomisation, 509 patients were allocated to receive treatment, 491 to receive placebo. Using traditional statistical methods, absolute changes in the outcome measure for the treatment group are compared with the placebo group. These results are illustrated with Forest plots, including subgroup analyses, in Fig 2. This traditional style of reporting of the outcome for each intervention group, including the estimated effect sizes and associated precision, aligns with consensus recommendations for presenting primary RCT outcomes, as outlined in the CONSORT statement [24].

There is a significant improvement in the outcome measure in patients who received the treatment compared with placebo (mean change 4.23, 95% CI 3.64 to 4.82). All subgroups, defined by the seven phenotypic variables, showed improved outcomes with the treatment compared with placebo. The size of treatment-related benefit varied according to the values of the critical response-determining variables (X, Y and Z), but not others (age, sex, V1 and V2) (Fig 2).

If we assume that changes in the outcome measure of 5 or above are clinically meaningful, the RCT evidence estimates that the number needed to treat (NNT) to achieve this clinically meaningful benefit is 2.62. This favourable NNT suggests that the new treatment is an effective one, although it would be important to weigh its benefits against any potential adverse effects, not modelled here. Importantly, based upon the results of this RCT, the new treatment appears to be an efficacious option for all patient groups.

### Analysis of the same data using machine learning

Despite the apparently positive RCT results, the ground truth is that 563 of the 1000 patients are not responsive to treatment (Fig 1). Clinical application of the RCT-based evidence from the previous section (in addition to further RCTs conducted) could result in 56.3% of patients being prescribed a treatment from which they receive no benefit, while being exposed to any side effects and risks associated with the treatment. This 56.3% of patients are unknowingly disadvantaged by the RCT analysis based on traditional statistics. These disadvantaged patients disproportionately include those with lower X values (non-responsive patient mean 70.3, SD 12.1 versus responsive patient mean 79.8, SD 15.2, p for difference <0.0001), higher Y values (non-responsive patient mean 71.1, SD 20.1 versus responsive patient mean 68.6, SD 12.5, p = 0.02) and patients without Z (proportion of non-responsive patients with Z 0.255, SD 0.43 versus responsive patients 0.817, SD 0.39, p < 0.0001). Therefore, even with a perfect collection of all the phenotypic variables necessary to determine which patients are treatment responsive, most patients nonetheless receive ineffective treatment, with systematic disadvantage to patients with low X values, high Y values or those for whom Z is not present.

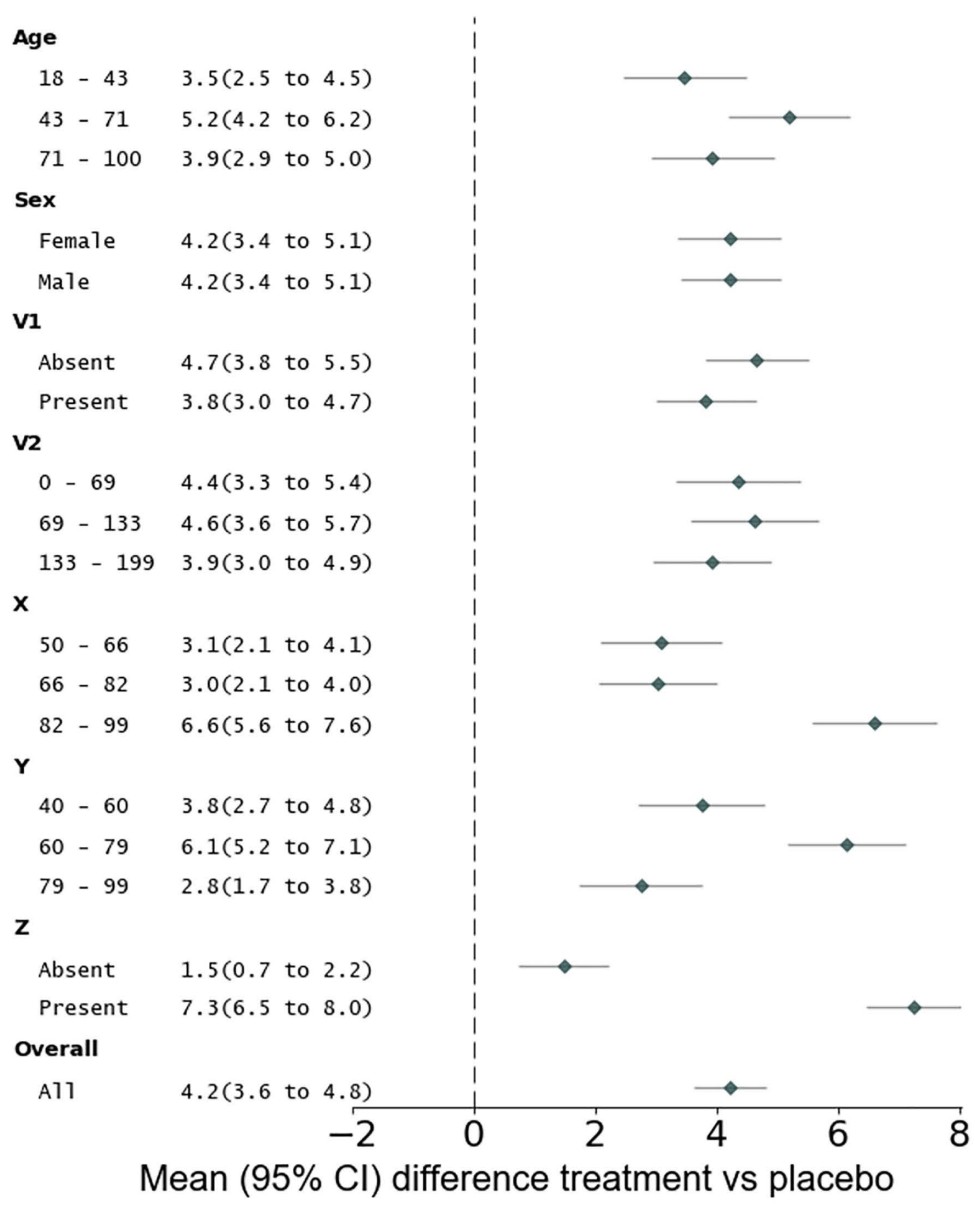

**Fig 2. Forest plots illustrating mean and 95% confidence interval change in the outcome measure in patients receiving treatment compared with placebo.** The bottom plot shows results across the entire cohort, with different subgroups plotted above that. CI = confidence interval.

ML analysis could help reveal insights which traditional analysis methods do not. Using the exact same simulated data as in the traditional RCT analysis described above, we next examined the capabilities of XGB ML analysis.

We first used XGB to identify whether it is possible to predict which patients benefit from treatment based upon their clinical phenotype, and who should avoid ineffective treatment. (See Methods for full details including five-fold cross-validation and XGB model parameters.) This analysis included only patients allocated to the treatment arm. We defined treatment responsiveness as an outcome measure of 5 or above, and non-response as values below 5. Using simulated data allowed us to compare the XGB predictions with the ground truth treatment response, in addition to the trial's

outcome measure, which may not always reflect true responsiveness. This comparison of XGB predictions with ground truth helps assess the generalisability of predictions beyond this individual trial's outcome data. Importantly, model training relied solely on data available to investigators, with the ground truth used exclusively for evaluation of prediction accuracy, not training.

Fig 3 shows the classification performance metrics for the XGB analysis compared with trial outcomes and ground truth responsivity. The XGB analysis predictions were 92.5% (95% CI 90.3–94.8) accurate according to the trial outcome data (Fig 3A) and 97.8% (95% CI 96.6–99.1) accurate according to ground truth (Fig 3B). The fact that XGB demonstrated higher accuracy compared to ground truth labels than the trial's outcome measure suggests that the model did not overfit to the noisy surrogate outcome data. Rather, XGB identified patterns in the underlying clinical phenotype data which offer generalisable insights that more accurately reflect the true treatment response. A power analysis revealed that the method was highly robust and a success criterion of >90% accuracy against ground truth was met in 100% of 500 simulation runs, resulting in a statistical power of 100% to detect the simulated effect under these conditions.

We also considered another commonly used form of ML analysis, logistic regression (LR). Using LR, predictions were 78.4% (95% CI 74.8–82.0) accurate according to the trial outcome data and 82.1% (78.8–85.5) accurate according to ground truth. The lower accuracy of LR compared to XGB is commonly found for non-linear multivariate interactions, such as in this analysis [17].

Even with perfect measurement of key variables, RCT analyses can miss crucial patterns. Conclusions drawn from traditional RCT analysis in the previous section could lead to 43.7% of patients benefitting from treatment, with the remaining majority of patients receiving an ineffective treatment. In contrast, XGB analysis of data from only half as many patients (specifically, those in the treatment group rather than the placebo group, which provides no insight into treatment responses) accurately predicted patients being treatment responsive or non-responsive in 97.8% of cases.

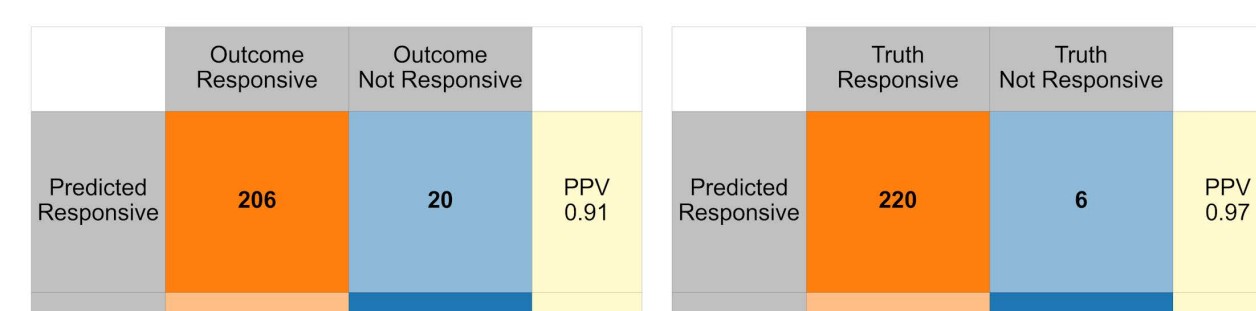

**A**  XGB predictions vs Trial outcome

| | Outcome Responsive | Outcome Not Responsive | |
|---|---|---|---|
| Predicted Responsive | **206** | **20** | PPV 0.91 |
| Predicted Not Responsive | **18** | **265** | NPV 0.94 |
| | Sensitivity 0.92 | Specificity 0.93 | 92.5% accurate |

**B**  XGB predictions vs Ground truth

| | Truth Responsive | Truth Not Responsive | |
|---|---|---|---|
| Predicted Responsive | **220** | **6** | PPV 0.97 |
| Predicted Not Responsive | **5** | **278** | NPV 0.98 |
| | Sensitivity 0.98 | Specificity 0.98 | 97.8% accurate |

**Fig 3. Confusion matrices with classification metrics for predictions of treatment response using XGB.** Predictions of treatment response using XGB analysis are compared with the treatment response apparent in the trial outcome measure (A) and the ground truth **(B)**. Orange shading denotes treatment responsive (suggested in the trial outcome or from ground truth) and blue shading denotes non-treatment responsive cells; bold orange/blue denote correct treatment allocation according to the outcome; light orange/blue denote inappropriate treatment allocation. PPV: positive predictive value; NPV: negative predictive value.

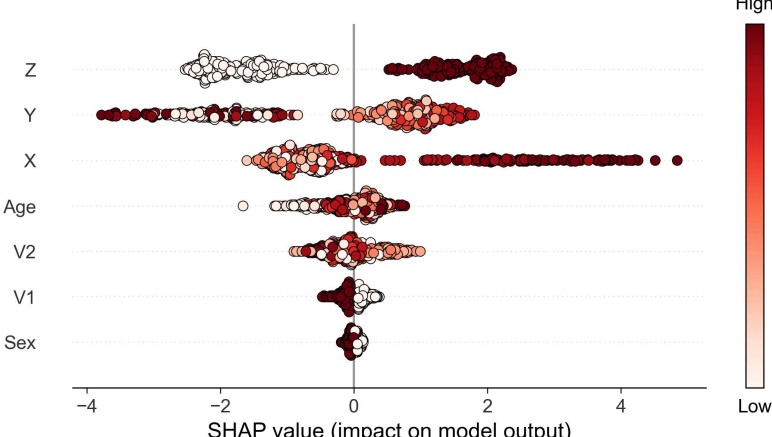

**Fig 4. SHAP (SHapley Additive exPlanations) values for each feature in the ML model.** Each point on the plot represents a variable's SHAP value for each of the 509 patients receiving treatment for whom predictions of treatment response were made. Colours represent the value of the variable from low (white) to high (dark red). Variables are ordered by their impact on model outputs, from highest (top) to lowest (bottom), based on the sum of SHAP value magnitudes across all predictions.

## Additional benefits of a machine learning analysis

The XGB approach enables interrogation of the model using SHAP (SHapley Additive exPlanations) values, to identify phenotypic variables which are influential in determining the model's predictions about treatment response [25]. Here, each SHAP value indicates how much an input variable contributed to the model's prediction of each patient's treatment response, shown in Fig 4. A large positive SHAP value indicates an influence for predictions of treatment responsive, whereas a large negative value indicates an influence for predictions of non-responsive. The colour of the plots indicates the value of the associated variable: dark red represents higher values of a continuous variable or the presence of a binary variable, while white indicates lower values of the continuous variable or the absence of the binary variable. All seven phenotypic variables are ranked from highest to lowest influence on model predictions, based on the total magnitude of SHAP values across all predictions. This ranking reflects the relative importance of each variable in influencing the model's decisions. We see that the three true response-determining variables – X, Y and Z – are identified as being the ones which most strongly influence the model outputs (Fig 4).

From Fig 4, we also observe that certain values of the X variable have a substantial impact on the model's predictions of treatment responsiveness, as evidenced by large positive SHAP values. The dark red shading of these points indicates that high values of X have large impact on the model predicting a patient as being treatment responsive. To explore this relationship further, in Fig 5, we plotted X values (x-axis) for each patient against their corresponding SHAP values (y-axis), coloured by the XGB prediction (orange for treatment-responsive, blue for non-responsive). Overlaid histograms show a summary of the proportion of responsive (orange) and non-responsive (red) predictions for the corresponding X values. Fig 5A shows that X values greater than 90 are associated with very high SHAP values, indicating a strong influence on the model's output being that the patient is treatment responsive. Specifically, X values above 95 almost always predict treatment responsiveness, while those between 90 and 95 still frequently result in a prediction of treatment responsiveness. This demonstrates that X values above 90 are highly influential in predicting treatment responsiveness. Conversely, when X values are below 90, their impact on the model's predictions is diminished.

Fig 4 also reveals that certain Y values significantly affect predictions of non-treatment responsiveness, as indicated by large negative SHAP values. Fig 5B examines the role of Y values when X is not influential (i.e., X is below 90). It

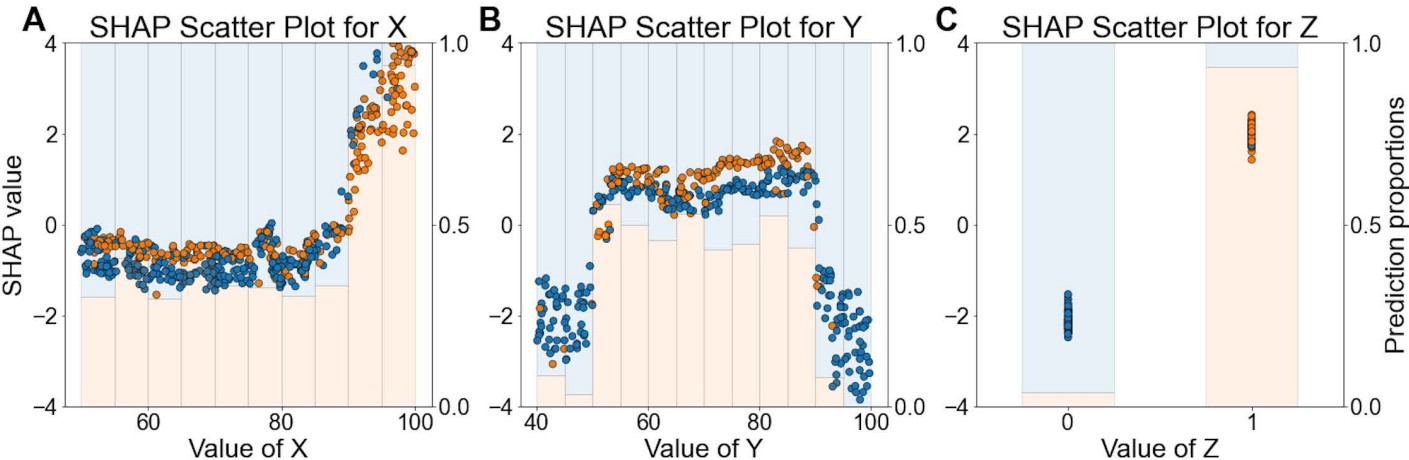

**Fig 5. Scatter plots of variable values versus SHAP values to interpret how specific variable ranges influence treatment predictions. A: Scatter plot of X values on the x-axis versus SHAP values (left y-axis).** SHAP values reflect the importance of each X value in predicting treatment response, with higher positive values indicating greater importance for predicting treatment responsiveness. The colour of plots indicates what the prediction was, with orange indicating a prediction of treatment responsive, and blue non-responsive. Overlayed histograms show the proportion of predictions (right y-axis) for different X values: orange bars denote treatment-responsive predictions, and blue bars denote non-treatment-responsive predictions. B: Scatter plot of Y values on the x-axis versus SHAP values, filtered to include only instances where X is below 90. This plot demonstrates the importance of Y values in the model's predictions, where lower negative SHAP values suggest higher importance for predicting non-responsive to treatment. Histograms overlayed on the scatter plot represent the proportion of predictions for different Y values, with orange bars for treatment-responsive and blue bars for non-treatment-responsive predictions. C: Scatter plot of Z values on the x-axis versus SHAP values, with data filtered to include only Y values between 50 and 90, as well as X values below 90. SHAP values indicate the importance of Z values in the model's predictions, considering the constraints on X and Y. The histograms show the proportion of predictions for Z values, with blue bars representing treatment-responsive predictions and blue bars representing non-treatment-responsive predictions.

shows that Y values below 50 or above 90 are linked to large negative SHAP values, suggesting these Y values are strong predictors of non-responsiveness in cases where X is below 90. Consequently, most predictions in these scenarios are for non-responsiveness. Additionally, Fig 4 indicates that Z generally has a significant influence on model predictions. Fig 5C focuses on cases where X is below 90 and Y is between 50 and 90. It shows that the presence of Z is associated with high positive SHAP values, correlating with a high likelihood of treatment responsiveness. Conversely, the absence of Z is associated with low negative SHAP values, indicating non-responsiveness. Thus when X is below 90 and Y is between 50 and 90, Z's presence predicts treatment responsiveness, while its absence suggests non-responsiveness.

In summary:

• X values above 90, especially those above 95, are strong predictors of treatment responsiveness.

• When X is below 90, Y values below 50 or above 90 suggest non-responsiveness.

• For X below 90 and Y between 50 and 90, Z's presence indicates treatment responsiveness, whereas its absence indicates non-responsiveness.

This analysis clarifies how X, Y, and Z values contribute to the model's predictions of treatment response. All these values align closely with the ground truth (see Fig 1).

With XGB, it is straightforward to identify key variables contributing to the model's accurate predictions of treatment response, as well as important values of these variables.

## Impact of data deficiency upon the machine learning analysis

We next considered an alternative scenario, which is identical to the original RCT, in the same patients, except that one response-determining clinical variable, Z, was not collected by the investigators. With this single data deficiency, the accuracies of the XGB model's predictions were 68.8% (95% CI 64.7–72.8) and 69.4% (95% CI 65.3–73.4) relative to trial outcomes and ground truth, respectively (S1 Fig).

In this scenario, the XGB predictions still outperform treatment recommendations drawn from traditional RCT-based analysis (i.e., where only 43.7% of patients receive appropriate treatment). However, a single piece of missing clinical information results in far less accurate predictions by the XGB model compared with analysis with complete information (Fig 3). This finding highlights the important of comprehensive clinical data collection to realise the potential of ML analysis techniques. If the binary variable Z was not accounted for and characterised during data collection, investigators will inevitably be blind to its importance, regardless of the sophistication of the analytical methods used.

## Impact of data excess upon the machine learning analysis

To address the problem of potential data deficiency, collection of additional variables is necessary. However, this solution must be balanced against the potential drawbacks of data *excess*. To evaluate the potential disadvantages of gathering more clinical data, we finally considered scenarios in which many more variables are collected and incorporated into the analysis.

In our previous RCT scenarios, only three of the seven clinical variables (X, Y, Z) were meaningful for understanding treatment response, while the other four (Age, Sex, V1, V2) were 'noise', that XGB models had to 'filter' from the true 'signal'. We conducted similar XGB analyses, but in addition to these seven clinical variables, we progressively introduced further noisy variables, to evaluate the ability of XGB analysis to accurately detect true signals amidst increasing noise.

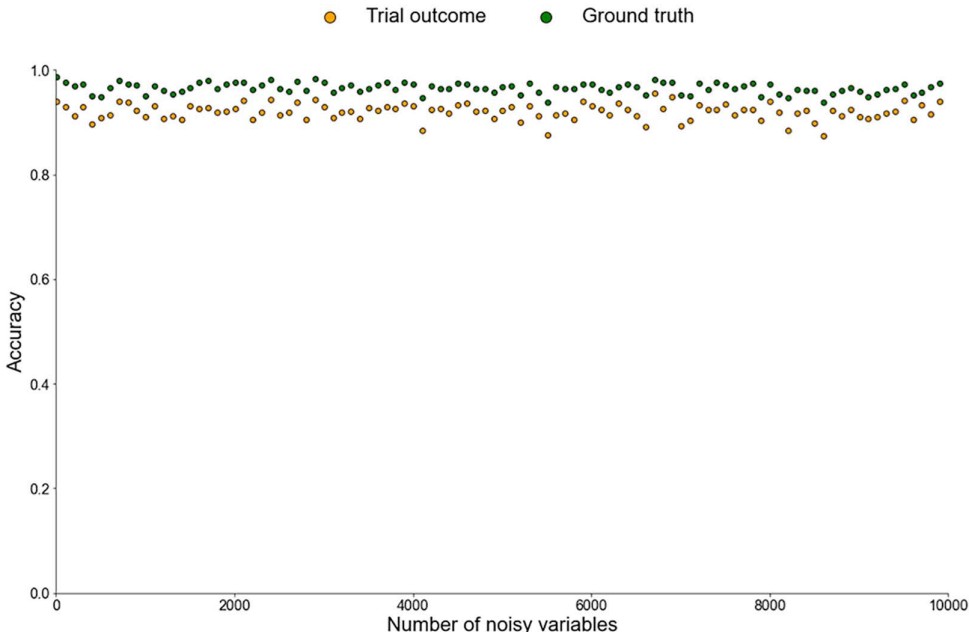

**Fig 6. Scatter plot depicting the accuracy of XGB treatment response predictions in relation to the number of noisy variables added to the original seven clinical variables.** Orange plots indicate accuracy compared with trial outcomes, while green plots indicate accuracy based upon ground truth.

These noisy variables were generated as a mix of binary and continuous types, with some covarying with X, Y or Z, while others remained completely independent of the existing variables. (For full details, see Methods). While all other aspects of the analysis approach and XGB model parameters remained unchanged, the clinical cohort was resampled to include these new noisy variables.

Fig 6 shows the accuracy of XGB treatment response predictions (based upon trial outcomes, in orange, and compared to ground truth, in green) as the number of additional noisy variables increases up to 10,000. Despite the increasing noise, XGB analysis consistently maintained high classification accuracy, with no observable decline in accuracy up to 10,000 variables.

The XGB analysis predictions again more closely mirrored ground truth treatment response, indicating they identified generalisable patterns rather than overfitting to a larger amount of noisy uninformative information.

With appropriate use of an XGB classifier, even the addition of up to 10,000 noisy clinical variables did not reduce classification accuracy, create spurious associations, or obscure true underlying effects.

## Discussion

Our analyses highlight two crucial points for clinical research: the potential benefits of ML analysis over traditional analysis approaches, and the necessity for comprehensive clinical phenotyping, to fully realise these benefits.

Crucial insights can be missed on account of how data are analysed. XGB detected critical information about the relationship between clinical phenotype and treatment response, information that traditional inferential methods had overlooked. Conclusions drawn from standard analysis of our simulated RCT could lead to just 43.7% of patients receiving appropriate treatment, and incorrect treatment would systematically disadvantage specific patient groups (those with low X, high Y, or absent Z). Subgroup analyses can identify broad differences in the primary outcome effect sizes based on individual clinical variables. However, as shown in Fig 2, these analyses may still indicate significant improvements compared to baseline when there is heterogeneity within the subgroups or multivariate relationships. Just because we cannot detect discrimination, it does not mean it is not there; the problem of "unknown unknowns" persists. In contrast, XGB analysis of the same RCT data correctly predicted treatment response in 97.8% of patients, removing the discrimination of these patient groups.

Not all forms of ML may fully capture complex relationships. Here, we observed that LR had lower accuracy (82.1%) than XGB. LR may fail to detect nonlinearities, which likely accounts for its reduced accuracy in predicting treatment responses [17], highlighting that it is not simply a matter of "ML is good, non-ML is bad". Indeed, inferential statistics offers several advantages over ML, such as providing clearer insights into causality and being more robust to overfitting [17]. Our recommendation is to integrate traditional inferential statistics from RCTs with ML analysis tailored to the data and specific research question(s). In many circumstances, ML approaches may not outperform traditional methods, especially if treatment response is related to simple, univariate and/or linear trends; if datasets are very small where complex models risk overfitting; or when strong theoretical knowledge already provides a robust explanatory framework. However, while RCT statistics provide valuable insights in many circumstances, ML can uncover more complex, multivariate, and non-linear patterns within the data that standard methods may overlook, thereby maximising the research value derived from clinical datasets.

To fully realise the benefits of ML analysis for patients, clinical phenotypes need to be characterised comprehensively. Here, the exclusion or inclusion of a single phenotypic variable (Z) determined whether the accuracy of XGB treatment predictions was 69.4% or 97.8%, respectively. This 28.4% difference is larger than the 19.4% difference between random chance predictions (50%) and using ML for analysis of incompletely characterised clinical phenotypes (69.4%). This underscores the need for "deep phenotyping", capturing detailed and diverse clinical data [26–28].

Clinical phenotypes are the most direct indicator of how disease is manifesting in patients. Many conditions sharing a diagnostic label are formed of a highly heterogeneous set of underlying pathologies. Treatment options for these

conditions can be similarly diverse. For example, a range of autoimmune conditions are treated with various combinations of steroids, intravenous immunoglobulins, plasma exchange, monoclonal antibodies, methotrexate, azathioprine and mycophenolate mofetil. Similar diversity in disease processes and available treatment options is present for conditions like epilepsy and headache. In clinical practice, we collect detailed information to understand how, when, where, and why different symptom combinations manifest, which ultimately guides patient care. However, this level of detail is often lacking in research settings, where there is instead a tendency to simplify and isolate a few factors, often with strict exclusion criteria and randomisation, to minimise the influence of comorbidities and other variables. The precise phenotypic variables which might be critical will vary dependent upon the specific circumstances of an investigation, but for one specific example, highly impactful and rigorous trials in stroke medicine often reduce a complex clinical syndrome to a quantified NIH Stroke Scale (NIHSS) score, time since symptom onset, and a limited number of comorbidities such as diabetes, atrial fibrillation, and hypertension [29–31]. However, critical nuances in the clinical history, such as previous recurrent episodes which might indicate cerebral amyloid angiopathy, which can increase the risk of haemorrhagic complications from thrombolysis, may be overlooked in initial trials and can take several years to become apparent [32].

Collecting and analysing detailed patient data has traditionally presented challenges for clinical research. Using univariate statistical analysis, multiple comparisons across subgroups increases the risk of identifying spurious associations, and necessitates adjustments to statistical significance thresholds such as Bonferroni correction or false discovery rater control, which can obscure genuine trends [33]. Consequently, interventional studies using traditional inferential methods often limit data collection and/or analysis to a small subset of variables to verify the similarity between treatment and control groups at baseline. While this approach helps control confounding factors, it can produce findings that are less representative of the broader clinical population, and resulting evidence on the most effective treatments for specific patient subgroups is, therefore, limited.

ML analysis can instead allow variance across diverse clinical features to be embraced, studied and understood. The XGB analysis effectively identified and filtered relevant variables, providing insights which more accurately reflected ground truth than the imperfect surrogate trial outcome measure. Even in the presence of 10,000 additional pieces of noisy, covarying clinical information, the XGB analysis remained sensitive to true trends, without making spurious associations.

Collecting more detailed phenotype data from existing patients can also be less resource-intensive than recruiting additional patients. Recruiting a single new patient involves multiple time-consuming steps, including consent, eligibility screening, assessments, treatment, monitoring, and follow-up. In contrast, gathering additional phenotypic variables for each patient potentially takes seconds, or minutes, even across an entire cohort. If these variables are extracted automatically from an electronic health record (EHR), it may require no additional researcher time at all.

Advances in natural language processing techniques enable large-scale extraction of clinical information from EHRs [34–37], and there is a growing use of information captured during routine clinical care and stored within EHRs for research [38–41]. Utilising data from EHRs and routine clinical practice could allow researchers to compare different management strategies with outcomes in patient populations that better reflect real-world settings, providing larger-scale insights at lower cost. The most accurate reflection of a broad clinical population is the clinical population itself. However, the feasibility and insights possible from this form of research will entirely depend upon how comprehensively phenotypes and clinical outcomes are documented in routine care.

ML analysis can also help inform whether a more detailed clinical phenotype is required. In our analysis, the much lower accuracy of XGB predictions when a single, key, variable was missing, indicates that treatment effects are not fully understood, and gathering more phenotypic information could be beneficial. Conversely, very high prediction accuracy indicates that treatment responses are well understood, and that further data collection might yield diminishing returns, suggesting that research efforts could be more effectively focused elsewhere. With traditional inferential analysis, the same missing information does not directly affect the primary outcome measure; the analysis is "blind" to the existence and impact of such "data gaps".

A common concern with ML models is their lack of interpretability, which can impede clinicians' understanding and trust in their outputs, limiting adoption into clinical practice. However, we have demonstrated that thorough interrogation of a trained XGB ML model can reveal which variables, and their specific values, determine clinical treatment response. This information could aid in the development of treatment algorithms similar to those used for mitigating cardiovascular disease risk, which take into account specific ranges of values for blood pressure, cholesterol levels, weight, height, and various other variables [42]. This interrogative approach to ML can highlight critical aspects of a disease that warrant further attention, in clinical practice, research, and model development.

We advocate that a post-hoc data-driven ML analysis, such as those described here, should become a routine part of any clinical research study with a suitably detailed and large dataset. While XGB can function effectively with smaller datasets of a few hundred patients, as is illustrated here, its predictive power is enhanced with larger and more diverse datasets. As the number of clinical phenotype variables increases, the 'curse of dimensionality' demands more patients to ensure reliable results, as the data required for accurate generalisation grows exponentially [43,44]. XGB typically performs best with thousands to hundreds of thousands of samples, enabling it to capture complex relationships and nuances within the data [18,45]. This approach could help reveal novel hypothesis generating information and lead to clinically meaningful insights which might otherwise have been missed and taken decades to eventually become apparent.

ML analysis need not be complex or resource intensive. All the analyses presented here were run within a minute on a single desktop computer. Our analyses serve as a clear example of how a straightforward gradient boosting ML classifier (XGB) can be effectively used in clinical research to achieve high predictive performance, versatility, scalability, and interpretability, to produce generalisable results. While this specific example used a binary classification predicting treatment response, the XGB algorithm can be easily adapted to perform regression, ranking, or other forms of prediction [18]. Applying these techniques effectively in practice requires systematic hyperparameter tuning (e.g., grid search, random search, Bayesian optimisation, LR elastic models) and sensitivity analysis to refine model performance. Although a detailed exploration is outside this work's scope, practical guidance on these techniques is readily available in other resources [46,47].

Quality evaluations of clinical trials should arguably place greater emphasis on both how data are analysed and the extent of clinical phenotyping, the two topics we examined here. The purpose of evaluating trial quality is to establish their ability to reliably identify key information. Our findings clearly indicate that both the analysis methods used and the extent of clinical phenotyping directly influences a study's ability to identify such key information. Various frameworks exist to evaluate clinical trial quality [24,48–51]. Assessments currently centre around details such as randomisation, blinding, follow-up, analysis, and reporting of a pre-registered primary outcome. There is emphasis on ensuring patient sample sizes are sufficiently large to produce accurate estimates of population averages. As the number of patients in a study increases, the ability to detect smaller effect sizes improves, allowing identification of subtle associations between variables. However, once an appropriately powered sample size is reached, further increases yield diminishing returns. In our simulations, expanding the sample size to millions of patients would not have significantly changed the conclusions, but a single missing piece of clinical phenotype information greatly limited the insights gleaned. Rather than considering whether this type of insight is being missed, current evaluation frameworks generally focus on clinical data collection only as a means to confirm treatment arm similarity at baseline [50], for reporting of baseline characteristics, or ancillary analyses including subgroup evaluations [24]. Our traditional RCT analysis was conducted in a manner which could score "full marks" on relevant current evaluation metrics. However, without using XGB analysis and capturing Z, the treatment recommendations from the RCT are a poor representation of the ground truth, causing the majority of patients to receive inappropriate treatment. Current trial evaluation quality criteria credit a study's ability to detect very small effect sizes more than the ability to detect much larger, impactful, effects demonstrated here.

Another key aspect of evaluating trial quality is identifying and minimising potential sources of bias. We have already discussed how, in our analysis, traditional inferential methods resulted in systematic discrimination against certain patient

groups. Limiting the breadth of data collection can introduce bias by systematically disadvantaging certain patient groups, and the omission of important information may lead to trial outcomes that misrepresent these patients in direct proportion to how much they deviate from the mean of the uncollected data [52].

In clinical situations that appear to have a simple binary endpoint, such as survival or complete resolution of symptoms from an acute infection, a ML model's utility can come from considering more granular, secondary outcomes that capture the patient's recovery trajectory. For example, even when a treatment ensures patient survival, there is often significant heterogeneity in the time to symptomatic resolution, the incidence of treatment-related adverse events, or the duration of hospitalisation. By defining the 'treatment effect' in terms of these more nuanced metrics, ML analysis can still identify patient phenotypes associated with more or less favourable responses. The approach's core strength lies in potentially helping explain any form of clinically relevant, treatment-related variance.

This work should be viewed as an important proof of principle and baseline upon which greater complexity can be built. We used a single, simplified scenario as a starting point, but it has limitations. We modelled data as complete, though missing data is common in real-world scenarios. However, XGB models can handle missing data effectively. ML models, particularly with high-dimensional data, risk overfitting, which can limit generalisability [16]. To address this, we tuned XGBoost parameters to reduce overfitting while preserving accuracy, ensuring robust, generalisable trends. Using a tree depth of three prioritises simplicity and interpretability, while avoiding overfitting and ensuring meaningful, low-order interactions can be captured. This shallow structure establishes a baseline against which future iterations can be compared, where deeper trees or more complex models could explore higher-dimensional, non-linear relationships if needed. Smaller cohorts and effect sizes could make it challenging for ML to reliably detect trends, so ML analysis is not suitable for all clinical research studies. This work demonstrates what can be achieved with a few hundred well-characterised cases, rather than the tens of thousands of data points required for some more complex, computationally intensive ML analyses. While this study focused on a single primary endpoint at one time point, ML has the potential to incorporate more nuanced analyses, such as time-course data.

ML techniques can deliver highly accurate, explainable, and generalisable predictions by analysing intricate interactions among multiple clinical variables. When appropriate analysis techniques are used, every piece of new clinical information collected has the potential to unlock new understanding of a disease which could benefit patients. There should be a greater drive to more comprehensively capture how diseases manifest with better clinical data, to enable patients to benefit from the potential insights which ML makes possible.

## Methods

### Creation of simulated clinical cohort data and determinants of 'ground truth' treatment response in a randomised control trial

For each member of the simulated clinical cohort of 1000 patients, we generated a set of arbitrary clinical phenotype variables. To minimise assumptions about the data's underlying distributions, each variable was modelled independently and drawn from uniform distributions:

- Age – an integer uniformly distributed between 18 and 100.

- Sex – a binary variable with equal probability for the outcomes 0 and 1.

- V1 – a binary variable with equal probability for the outcomes 0 and 1.

- V2 – an integer uniformly distributed between 0 and 200.

- X – an integer uniformly distributed between 50 and 100.

- Y – an integer uniformly distributed between 40 and 100.

- Z – a binary variable with equal probability for the outcomes 0 and 1.

These variables represented both binary and continuous data to reflect heterogeneous types of data used in clinical research. The lower and upper bounds for each variable were selected arbitrarily.

Whether or not a patient would be responsive to the treatment was determined by a simulated non-linear multivariable relationship between X, Y and Z illustrated in Fig 1 and explained in the associated text of the results section. In summary, patients were responsive to the treatment if X was 95 or above; if Z was present and Y was between 50 and 90; if Y was between 50 and 90 and X was between 90 and 95.

For the simulated RCT, all members of the cohort were randomly assigned in a 1:1 ratio to receive either the treatment or placebo. Trial outcome data were then drawn from either of two distributions:

- A 'responsive' distribution – for patients receiving treatment who are treatment responsive, where the outcome measure change was drawn from a distribution with mean 10 and standard deviation 3.

- A 'non-responsive' distribution – for patients receiving placebo or treatment which they are not responsive to, where the trial outcome measure change was drawn from a distribution with mean 0 and standard deviation 3.

For the analysis considering data deficiency, the Z variable was omitted from the data used for model training in the XGB analysis. All other values and variables were kept unchanged.

For the analysis considering data excess, additional 'noisy' variables were generated, starting with 5 noisy variables and increasing by 100 at each step, up to a total of 10,000 variables. One-sixth of the noisy variables were continuous and covaried with X. This covariance was established by taking the patient's X value and adding a noise signal randomly drawn from a normal distribution with a mean of 0 and a standard deviation of 10. Another sixth of the noisy variables were continuous and covaried with Y, following the same method. Additionally, another sixth of the noisy variables were binary and covaried with Z, where the covariance was defined as a random selection of two-thirds of the values matching the patient's Z value. The remaining half of the noisy variables were independent of all other variables, comprising either binary variables (randomly drawn from a probability distribution where the variable was present in 10% of patients and absent in 90%) or continuous variables (random integers between 18 and 100). The clinical cohort was resampled for this analysis, using the exact same distributions and definitions as above.

## Analysis of the randomised control trial data using traditional inferential statistics

All statistical analyses illustrated in Fig 2 were performed using the NumPy toolbox [53] in Python version 3.11.5. The sample size of 1000 patients provides 80% power to detect an effect size of 1.064. All outcome end points are reported in terms of the mean absolute change in the outcome variable and 95% confidence intervals for that change in the treatment group versus placebo. The subgroup confidence intervals have not been adjusted for multiple comparisons which limits the ability to infer definitive treatment effects based upon them in isolation. Forest plots were produced using the Python forestplot toolbox [54]. All mean and 95% confidence interval values are reported to 2 decimal places.

## Machine learning analyses

We conducted XGB ML analyses to examine the relationship between the simulated clinical phenotype variables and treatment responsiveness. The XGB models were trained to process all the variables known about each patient to make predictions about whether a patient is treatment responsive. In real life, clinical investigators would not be aware of ground truth treatment response and would have to rely upon trial surrogate trial outcome data. For all model training, we therefore used only the trial outcome as a surrogate of treatment response and did not include ground truth information. The XGB models were trained to predict whether patients had a change in the outcome variable or 5 or more based upon information about all the clinical phenotype variables. Ground truth information was only used later, in evaluation of predictions made using the trial outcome data.

Data preprocessing for all the XGB analyses used the same steps each time. Non-binary data were scaled by removing the mean and scaling to unit variance, to ensure that each variable contributed equally to the model's training.

An XGBoost binary classifier was initialised with a predefined set of hyperparameters [18]. The same model parameters were used in every analysis. The maximum depth of each tree was set to 3 to help prevent overfitting by limiting the complexity of the model. Learning rate was set to 0.1, this determines the step size at each iteration and the value was chosen to ensure a balance between model accuracy and computational efficiency. The number of the number of trees (or rounds) in the model was 100. The subsample parameter was set to 0.8, meaning that 80% of the training data was randomly sampled to grow each tree. Similarly, 80% of the features were randomly sampled when creating each tree, thus introducing randomness and reducing overfitting. These pre-defined model hyperparameters were chosen to ensure performance and reliability of the models in each of the XGB analyses and minimise risk of overfitting.

To ensure robust model evaluation, a five-fold cross-validation approach was implemented using the KFold method in all the XGB analyses. Out-of-fold predictions for each sample were obtained and these predictions on unseen data were used as the model prediction data. Predictions on the unseen data were compared with true treatment response (according to either trial outcome data values above/below five or ground truth) and sorted into confusion matrices (Fig 3 and S1 Fig) indicating the number of true positive (top left), false positive (top right), false negative (bottom left) and true negative (bottom right). Sensitivity, specificity, positive predictive value (PPV), and negative predictive value (NPV) were calculated for each classification. The overall accuracy of predictions is indicated in the bottom right corner of each confusion matrix. 95% confidence intervals for the overall accuracy estimates were calculated using the normal approximation of the binomial distribution.

To quantify the statistical power and robustness of our analytical approach, the entire data generation and XGB analysis pipeline was repeated for 500 independent runs. For each run, a "successful detection" was defined as the model achieving an accuracy greater than 90% when classifying against the known ground truth.

To interpret XGB model predictions, SHAP (SHapley Additive exPlanations) values were computed for each of these XGB model predictions to determine how much each variable had impacted each prediction [25]. The SHAP values were then used to generate a summary plot, visualizing feature importance with a dot plot (Fig 4).

Fig 4 revealed that variables X, Y, and Z were the most influential in determining model predictions. To understand which specific values of these key variables contributed to treatment response, we first examined X, which, in some instances, had a large impact on predictions, indicated by high positive SHAP values above 2 in Fig 4. We generated a scatter plot of X values on the x-axis and their corresponding SHAP values on the y-axis (Fig 5A). This plot showed that X had a less pronounced effect when its value was below 90. Next, we identified that certain Y values were associated with negative predictions, as indicated by SHAP values below −2 in Fig 4. To investigate further, we plotted Y values against SHAP values for all patients when X was not influential, i.e., X values below 90 (Fig 5B). This plot revealed that Y had a reduced impact when its values ranged between 50 and 90. Finally, for patients with X values below 90 and Y values between 50 and 90, we plotted Z values on the x-axis against SHAP values on the y-axis to explore Z's effect on predictions (Fig 5C).

For the LR analysis, default model parameters from the scikit-learn toolbox [47] were used. Data pre-processing followed the same steps as described above for the XGB analyses, with the exception that interaction terms between all combinations of variables were included for the LR model. This adjustment was made to account for potential interactions between the variables, ensuring the LR model's ability to accurately detect multivariate effects was not excessively compromised. L2 regularisation was applied to minimise overfitting. The optimization problem was solved using the 'lbfgs' algorithm. Tolerance for stopping criteria was set to 0.0001 and maximum number of iterations to 100.

## Supporting information

**S1 Fig. Confusion matrices with classification metrics for predictions of treatment response using XGB, when variable Z is removed from consideration.** Predictions of treatment response using XGB analysis are compared with the treatment response apparent in the trial outcome measure (A) and the ground truth (B). Orange shading denotes

treatment responsive (suggested in the trial outcome or from ground truth) and blue shading denotes non-treatment responsive cells; bold orange/blue denote correct treatment allocation according to the outcome; light orange/blue denote inappropriate treatment allocation. PPV: positive predictive value; NPV: negative predictive value.
(DOCX)

## Author contributions

**Conceptualization:** Stephen D. Auger, Gregory Scott.

**Data curation:** Stephen D. Auger.

**Formal analysis:** Stephen D. Auger.

**Investigation:** Stephen D. Auger.

**Methodology:** Stephen D. Auger.

**Resources:** Stephen D. Auger.

**Software:** Stephen D. Auger.

**Validation:** Stephen D. Auger.

**Visualization:** Stephen D. Auger.

**Writing – original draft:** Stephen D. Auger.

**Writing – review & editing:** Stephen D. Auger, Gregory Scott.

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
