## [Decision Letter · Decision Letter 0]

24 Apr 2025

Dear Dr. Auger,

Thank you for submitting your manuscript to PLOS ONE. After careful consideration, we feel that it has merit but does not fully meet PLOS ONE’s publication criteria as it currently stands. Therefore, we invite you to submit a revised version of the manuscript that addresses the points raised during the review process.

We look forward to receiving your revised manuscript.

Kind regards,

Matthew Chin Heng Chua

Academic Editor

PLOS ONE

“SDA is funded by a UK National Institute for Health and Care Research (NIHR) Clinical Lectureship. GS is funded by the National Institute for Health Research [Advanced Fellowship].

The funders had no role in study design, data collection and analysis, decision to publish, or preparation of the manuscript. The views expressed are those of the authors and not necessarily those of the NIHR or the Department of Health and Social Care.”

4. Please remove your figures from within your manuscript file, leaving only the individual TIFF/EPS image files, uploaded separately. These will be automatically included in the reviewers’ PDF.

Reviewers' comments:

Reviewer's Responses to Questions

**Comments to the Author**

1. Is the manuscript technically sound, and do the data support the conclusions?

Reviewer #1: Yes

2. Has the statistical analysis been performed appropriately and rigorously?

Reviewer #1: Yes

3. Have the authors made all data underlying the findings in their manuscript fully available?

Reviewer #1: Yes

4. Is the manuscript presented in an intelligible fashion and written in standard English?

Reviewer #1: Yes

Reviewer #1: In this work, a randomised clinical trial has been stimulated to compare gradient boosting (XG boost) machine learning with traditional analysis when “ground truth” treatment responsiveness depends on the interaction of multiple phenotypic variables. The detected results from traditional analysis indicated that outcome measure change from baseline was 4.23 (95% CI 3.64–4.82) of patients. In contrast, the treatment response of patients reached at 97.8% (95% CI 96.6–99.1) in the context of machine learning. Notably, analysis results drop to 69.4% (95% CI 65.3–73.4) because of an omitted single variable. These indicated that machine learning could be maximized insights derived from clinical research studies. Overall, it is meaningful work, but several points should be clarified.

1. The introduction did not sufficiently highlight the innovation and superiority of this work. The authors are encouraged to clarify the innovation and superiority of their work.

2. Clarify methodological generalizability and limitations of ml approaches. While the discussion provides a strong case for the superiority of XGB in this particular context, the manuscript would benefit from a more balanced and critical evaluation of the generalizability of these findings. For example, the authors could elaborate on the types of datasets or clinical conditions where ML approaches like XGB may not outperform traditional methods, as well as practical constraints such as data availability, model interpretability, or computational requirements. This would strengthen the manuscript by providing a more nuanced view of ML’s utility across diverse clinical settings.

3. Elaborate on the nature and impact of clinical phenotyping. The manuscript emphasizes the importance of “comprehensive clinical phenotyping,” but does not provide sufficient detail about what this entails in practice. The authors are encouraged to elaborate on the specific phenotypic variables critical to the model’s success and discuss how such detailed phenotyping can be realistically obtained in real-world clinical trials. Providing concrete examples or frameworks would enhance the translational relevance and practical guidance for future research.

4. The simulation assumes linear effects for non-critical variables (V1/V2), which may oversimplify real-world clinical complexity. For instance, age or biomarkers rarely exhibit uniform noise distributions. Introducing realistic correlations or nonlinear interactions among variables would enhance ecological validity. Additionally, XGBoost hyperparameters (e.g., max depth=3) are justified but lack sensitivity analysis. Reporting performance variation with deeper trees or alternative tuning strategies (e.g., grid search) would strengthen confidence in model robustness. Subgroup analyses in Figure 2 lack multiplicity adjustments, inflating Type I error risks. Acknowledging this limitation and proposing Bonferroni corrections or false discovery rate control would improve rigor.

5. The term "hidden treatment patterns" risks overstating novelty, as interactions are often hypothesized in clinical research. Clarify whether the "ground truth" represents simulated relationships or reflects established biological mechanisms (e.g., pharmacogenomics). Confusion matrices (Figure 3) use a colorblind-unfriendly red/green palette, risking misinterpretation. Replacing these with patterned fills or dual-labeling (e.g., text annotations) would improve accessibility. Reporting additional metrics like Cohen’s kappa or area under the receiver operating characteristic curve (AUROC) would contextualize accuracy improvements over chance expectations.

6. The study fails to connect findings to real-world clinical scenarios where phenotypic complexity limits trial generalizability (e.g., oncology, neurology). For example, how might missing variables like comorbidities or genomic markers affect ML predictions in diverse populations? Overstating ML’s ability to "maximize insights" without discussing computational costs, reproducibility challenges, or ethical implications (e.g., algorithmic bias in treatment allocation) undermines clinical utility. Underexplored potential biases (e.g., over-reliance on correlated variables like V1/V2) and the absence of temporal dynamics (longitudinal outcomes) limit translational relevance.

7. Claiming "data excess has no penalty" risks overgeneralization, as sparser signals or smaller datasets may suffer from noise-induced overfitting. Temper conclusions with caveats about scalability. The manuscript lacks engagement with existing literature on ML in clinical trials (e.g., SHAP/XGBoost applications in healthcare). A brief review contextualizing contributions (e.g., comparing to rule-based decision trees) would strengthen its novelty.

**Do you want your identity to be public for this peer review?** For information about this choice, including consent withdrawal, please see our Privacy Policy

Reviewer #1: **Yes: ** Xiaohai Zheng

---

## [Author Response · Author response to Decision Letter 1]

2 May 2025

Formatting has been amended to meet all the formatting requirements described within the links provided.

We have reviewed the policy and confirm all code will be shared in accordance with the policies.

“SDA is funded by a UK National Institute for Health and Care Research (NIHR) Clinical Lectureship. GS is funded by the National Institute for Health Research [Advanced Fellowship].

The funders had no role in study design, data collection and analysis, decision to publish, or preparation of the manuscript. The views expressed are those of the authors and not necessarily those of the NIHR or the Department of Health and Social Care.”

As advised, we have provided the amended Funding Statement in the Resubmission cover letter and removed the statement from Acknowledgements.

4. Please remove your figures from within your manuscript file, leaving only the individual TIFF/EPS image files, uploaded separately. These will be automatically included in the reviewers’ PDF.

We have removed all figures from the manuscript file.

We have added the Supporting Figure caption to the end of the Manuscript file as advised.

We thank the reviewer for their helpful comments and appreciate the chance to refine our manuscript. We have carefully considered each suggestion and below are our point-by-point responses, detailing the revisions incorporated into the manuscript based on the valuable feedback.

Reviewer #1: In this work, a randomised clinical trial has been stimulated to compare gradient boosting (XG boost) machine learning with traditional analysis when “ground truth” treatment responsiveness depends on the interaction of multiple phenotypic variables. The detected results from traditional analysis indicated that outcome measure change from baseline was 4.23 (95% CI 3.64–4.82) of patients. In contrast, the treatment response of patients reached at 97.8% (95% CI 96.6–99.1) in the context of machine learning. Notably, analysis results drop to 69.4% (95% CI 65.3–73.4) because of an omitted single variable. These indicated that machine learning could be maximized insights derived from clinical research studies. Overall, it is meaningful work, but several points should be clarified.

1. The introduction did not sufficiently highlight the innovation and superiority of this work. The authors are encouraged to clarify the innovation and superiority of their work.

We have now revised the Introduction (final paragraph) to explicitly highlight the core, novel aspect of the current work, as follows:

“In this study, we investigated how both the analysis approach and characterisation of clinical phenotype within an RCT influences our ability to infer underlying 'ground truth'. Using simulated data representative of typical RCTs, we conducted a novel, formal comparison contrasting the effectiveness of traditional statistical methods versus ML approaches at identifying the key factors and interactions driving treatment response.”

When coupled with the added context regarding existing clinical ML applications (addressing comment 7, below), we believe the manuscript now articulates its specific place and contribution within the field more clearly.

2. Clarify methodological generalizability and limitations of ml approaches. While the discussion provides a strong case for the superiority of XGB in this particular context, the manuscript would benefit from a more balanced and critical evaluation of the generalizability of these findings. For example, the authors could elaborate on the types of datasets or clinical conditions where ML approaches like XGB may not outperform traditional methods, as well as practical constraints such as data availability, model interpretability, or computational requirements. This would strengthen the manuscript by providing a more nuanced view of ML’s utility across diverse clinical settings.

We agree with the need for balanced discussion regarding the generalisability and limitations of ML approaches beyond these specific findings with XGB. In addition to the existing acknowledgement of the distinct strengths of traditional inferential statistics (paragraph 3, sentence 4), we have now explicitly incorporated the recommendation to elaborate on the types of datasets of clinical conditions where ML approaches may not outperform traditional methods (same laster in the same third paragraph):

“In many circumstances, ML approaches may not outperform traditional methods, especially if treatment response is related to simple, univariate and/or linear trends; if datasets are very small where complex models risk overfitting; or when strong theoretical knowledge already provides a robust explanatory framework.”

Regarding the importance of considering practical constraints, the Discussion addresses multiple key components of these practical considerations: challenges related to data availability and collection (Discussion paragraph 8), considerations surrounding model interpretability (Results section 4 and Discussion paragraph 11), and computational requirements (Discussion paragraph 13).

Together, these sections provide a balanced and comprehensive summary of some key considerations and limitations with ML relative to traditional inferential approaches.

3. Elaborate on the nature and impact of clinical phenotyping. The manuscript emphasizes the importance of “comprehensive clinical phenotyping,” but does not provide sufficient detail about what this entails in practice. The authors are encouraged to elaborate on the specific phenotypic variables critical to the model’s success and discuss how such detailed phenotyping can be realistically obtained in real-world clinical trials. Providing concrete examples or frameworks would enhance the translational relevance and practical guidance for future research.

We agree with the reviewer on the importance of clearly outlining what is meant by 'comprehensive clinical phenotyping’ in practice. To ensure this concept is clear from the outset, the second and third sentences of the Introduction orient the reader with an explanation of our use of the term 'clinical phenotype' and how these might be practically expanded to be more comprehensive .

Regarding the specific phenotypic variables critical to the model’s success in this study, there are two aspects to this: the predefined 'ground truth' variables and those identified empirically by the analysis. We hope the following locations clearly provide this detail: The 'ground truth' variables underpinning our simulations are presented visually in Figure 1 and described textually in the first sections of the Results and Methods. Subsequently, the variables identified as most important by the XGBoost analysis are presented and discussed in the Results section associated with Figures 4 and 5. Please let us know if there is any specific aspect which requires further explanation.

We acknowledge the practical challenges of acquiring detailed phenotypic data in real-world clinical trials. The specifics are highly dependent on the context of each study, making comprehensive coverage in a general discussion difficult. We address these practical considerations across several paragraphs in the Discussion (specifically 4, 5, 6, and 8), covering aspects such as utilising existing clinical records and the potential need for enhanced data capture structures. To provide a concrete example, as requested, paragraph 5 includes a description using stroke trials as an example comparing a relatively reductive NIHSS score with nuanced information which might be available within a clinical history (e.g., that which might indicate the presence of potential cerebral amyloid angiopathy) and impact treatment outcomes. Prompted by the reviewer's feedback, we have added clearer emphasis in paragraph 5:

“The precise phenotypic variables which might be critical will vary dependent upon the specific circumstances of an investigation, but for one specific example, highly impactful and rigorous trials in stroke medicine often reduce a complex clinical syndrome to a quantified NIH Stroke Scale (NIHSS) score, time since symptom onset, and a limited number of comorbidities such as diabetes, atrial fibrillation, and hypertension. However, critical nuances in the clinical history, such as previous recurrent episodes which might indicate cerebral amyloid angiopathy, which can increase the risk of haemorrhagic complications from thrombolysis, may be overlooked in initial trials and can take several years to become apparent.”

4. The simulation assumes linear effects for non-critical variables (V1/V2), which may oversimplify real-world clinical complexity. For instance, age or biomarkers rarely exhibit uniform noise distributions. Introducing realistic correlations or nonlinear interactions among variables would enhance ecological validity. Additionally, XGBoost hyperparameters (e.g., max depth=3) are justified but lack sensitivity analysis. Reporting performance variation with deeper trees or alternative tuning strategies (e.g., grid search) would strengthen confidence in model robustness. Subgroup analyses in Figure 2 lack multiplicity adjustments, inflating Type I error risks. Acknowledging this limitation and proposing Bonferroni corrections or false discovery rate control would improve rigor.

Regarding the reviewer's concerns about the potential oversimplification of assuming purely linear effects for the V1/V2 non-critical variables, we acknowledge that real-world clinical variables often exhibit complex, non-linear relationships and non-uniform noise distributions. While V1/V2 were kept simple for illustrative clarity, we designed the broader simulation involving up to 10,000 further non-critical variables specifically to address this concern by incorporating a variety of covariance structures, including both linear and non-linear interactions. We believe this approach significantly enhances the ecological validity beyond the baseline V1/V2 example. We recognise, as the reviewer implies, that comprehensively modelling all potential real-world biological complexity is an immense challenge, likely varying greatly between specific clinical scenarios. We believe the current simulation strikes a reasonable balance, providing a robust demonstration of the core concepts while acknowledging that specific applications would necessitate tailoring the modelled interactions to different specific contexts.

Our intention with the current manuscript was primarily to provide a proof-of-concept introduction to these ML methods for a predominantly clinical audience, hence our initial use of common default hyperparameters (like max depth = 3) to maintain focus on the core message. However, we are in full agreement that rigorous hyperparameter optimisation is a crucial step in developing deployable ML models. Therefore, following the reviewer's suggestion, we have now added text explaining this point to the Discussion paragraph 13:

“Applying these techniques effectively in practice requires systematic hyperparameter tuning (e.g., grid search, random search, Bayesian optimisation) and sensitivity analysis to refine model performance. Although a detailed exploration is outside this work's scope, practical guidance on these techniques is readily available in other resources.”

The new text is referenced to guide interested readers towards practical resources for exploring these optimisation strategies in greater depth.

Regarding multiplicity adjustments, we agree with the reviewer’s point, and this motivated our choice to present results in Figure 2 with only 95% confidence intervals rather than using significance of non-corrected statistical tests. We have fully incorporated the reviewer’s recommendation to propose Bonferroni corrections or false discovery rater control in the Discussion (paragraph 6):

Using univariate statistical analysis, multiple comparisons across subgroups increases the risk of identifying spurious associations, and necessitates adjustments to statistical significance thresholds such as Bonferroni correction or false discovery rater control, which can obscure genuine trends.

It is interesting to note that even with the potential inflated type I error risk, traditional methods still did not detect effects which are clear with ML analysis.

5. The term "hidden treatment patterns" risks overstating novelty, as interactions are often hypothesized in clinical research. Clarify whether the "ground truth" represents simulated relationships or reflects established biological mechanisms (e.g., pharmacogenomics). Confusion matrices (Figure 3) use a colorblind-unfriendly red/green palette, risking misinterpretation. Replacing these with patterned fills or dual-labeling (e.g., text annotations) would improve accessibility. Reporting additional metrics like Cohen’s kappa or area under the receiver operating characteristic curve (AUROC) would contextualize accuracy improvements over chance expectations.

Regarding use of the term ‘hidden treatment patterns’: We agree that interactions are often hypothesised in clinical research. However, our intention with this work is to consider the specific scenario where complex relationships that might not form part of a priori hypotheses might be uncovered through use of data-driven methods applied post hoc, as discussed in paragraph 12 of the Discussion.

Regarding clarification of what the ‘ground truth’ represents; we confirm this was indeed simulated to create a controlled environment for evaluating the methods. We have now added explicit statements clarifying this in both the Results (first section, paragraph 4):

“A simulated non-linear relationship between these three variables (X, Y, Z) determines which patients will be responsive to the treatment”

and Methods (section 1, paragraph 4):

“Whether or not a patient would be responsive to the treatment was determined by a simulated non-linear multivariable relationship”

Regarding metric reporting, we carefully considered the most appropriate metrics for illustrating the key points within the intended clinical context and opted to use confusion matrices. This choice was motivated by several factors: confusion matrices directly reflect performance at specific decision thresholds

---

## [Decision Letter · Decision Letter 1]

7 Aug 2025

Dear Dr. Auger,

Thank you for submitting your manuscript to PLOS ONE. After careful consideration, we feel that it has merit but does not fully meet PLOS ONE’s publication criteria as it currently stands. Therefore, we invite you to submit a revised version of the manuscript that addresses the points raised during the review process.

We look forward to receiving your revised manuscript.

Kind regards,

Ziheng Wang

Academic Editor

PLOS ONE

Journal Requirements:

Additional Editor Comments:

Reviewer 3 raises a concern regarding the lack of  the method section. However, this information appears to be included in the latter part of the manuscript. The authors may consider relocating or emphasizing this section earlier in the manuscript to ensure better visibility and avoid potential confusion for readers.

Reviewers' comments:

Reviewer's Responses to Questions

**Comments to the Author**

Reviewer #1: All comments have been addressed

Reviewer #2: All comments have been addressed

Reviewer #3: All comments have been addressed

Reviewer #4: All comments have been addressed

2. Is the manuscript technically sound, and do the data support the conclusions?

Reviewer #1: Yes

Reviewer #2: Partly

Reviewer #3: Partly

Reviewer #4: Yes

3. Has the statistical analysis been performed appropriately and rigorously?

Reviewer #1: Yes

Reviewer #2: No

Reviewer #3: Yes

Reviewer #4: Yes

4. Have the authors made all data underlying the findings in their manuscript fully available?

Reviewer #1: Yes

Reviewer #2: Yes

Reviewer #3: Yes

Reviewer #4: Yes

5. Is the manuscript presented in an intelligible fashion and written in standard English?

Reviewer #1: Yes

Reviewer #2: Yes

Reviewer #3: Yes

Reviewer #4: Yes

Reviewer #1: The authors have meticulously addressed all the concerns raised in the initial review. I recommend that this manuscript be accepted for publication in PLOS One.

Reviewer #2: The authors took the reviewers' comments into account.

However, the database simulation seems unrealistic.

First, it is impossible to have a database based only on gender and age as sociodemographic factors and on only five variables.

Furthermore, the age variable generally has a normal distribution. It is illogical to treat it as a uniform distribution and assign the same probability to a 100-year-old patient as to an 18-year-old patient.

How did the authors also choose the lower and upper bounds for each simulated variable?

We are in the era of open science, and there are certainly several databases in the context studied, drawn from the real world. Why didn't the authors consider concretizing their work by testing it on a real database (such as the dbGaP example)?

An important criterion is missing: model power

For the LR model, why was the choice of L2 penalized? Since this is a simulation, an LR elastic model should have been used to determine the exact values of L1 and L2 that are preferable according to the simulated cohort.

Several studies have validated the power of XGBoost to predict the phenomena studied on real databases. However, an interpretability problem remains compared to its recognized flexibility. This problem is also present in the simulation carried out in this article. What is the added value of this work?

It is important to note that the figures are of low resolution. Please increase it.

Reviewer #3: The methodology section used in this paper is missing. A well-developed and structured section is mandatory.

From the introduction section, we move on to the results section; without the methodology section, it is impossible to complete the article.

Result section:

Title (lines 106-108): Too long as a title

Fig 1. Investigation outline line133 :

for figure 1: what is this explanatory and interpretative paragraph? make the interpretations of your text then put figure 1 in parentheses at the end of the paragraph

title : An analysis of the randomised control trial data using traditional inferential statistics (lne 173 ) : Too long as a title

Fig 2 (line 185-187) : is't title ?!

The same remarks for the rest of the figures and paragraphs

Revise your text in the form of presenting the information so that it is easy for the reader to read

Reviewer #4: The revised manuscript "Machine learning detects hidden treatment response patterns only in the presence of comprehensive clinical phenotyping" is well written and an important contribution to the body of knowledge.

The authors responded well to all the criticisms and concerns raised by the reviewers and I feel the revised manuscript is fit for publication.

However, one minor observation is that the machine learning model as developed by the authors is generally more applicable to diseases without a definitive cause(s), for example degenerative conditions or chronic non-communicable diseases. I wonder how the model would be applied, for example, in case of an infection or infectious disease like malaria with a known cause that can be detected in a laboratory and used as a guide by clinicians in determining therapy? Because with most infections, once they are effectively treated, the symptoms like headache, vomiting, diarrhoea etc, disappear. In other words, can the authors provide a scope for the diseases where the model can best be applied?

**Do you want your identity to be public for this peer review?** For information about this choice, including consent withdrawal, please see our Privacy Policy

Reviewer #1: **Yes: ** Xiaohai Zheng

Reviewer #2: No

Reviewer #3: No

Reviewer #4: No

---

## [Author Response · Author response to Decision Letter 2]

11 Sep 2025

See also attached formatted document with full response to all reviewer comments.

Reviewer #1: The authors have meticulously addressed all the concerns raised in the initial review. I recommend that this manuscript be accepted for publication in PLOS One.

We thank the reviewer again for their previous comments. We are pleased they are satisfied with the amendments.

Reviewer #2: The authors took the reviewers' comments into account.

However, the database simulation seems unrealistic.

First, it is impossible to have a database based only on gender and age as sociodemographic factors and on only five variables.

Furthermore, the age variable generally has a normal distribution. It is illogical to treat it as a uniform distribution and assign the same probability to a 100-year-old patient as to an 18-year-old patient.

How did the authors also choose the lower and upper bounds for each simulated variable?

We are in the era of open science, and there are certainly several databases in the context studied, drawn from the real world. Why didn't the authors consider concretizing their work by testing it on a real database (such as the dbGaP example)?

We thank the reviewer for their comments.

Regarding choice of distributions and bounds, the variables were designed to be arbitrary placeholders to demonstrate a principle. We used uniform distributions for their simplicity and to make the fewest possible assumptions as a basis for generating random numbers. The variable labels "age" and "sex" were chosen purely to anchor the reader with familiar concepts for continuous and categorical data, respectively. We are not making any claims about clinical effects in specific age groups or modelling real-world demographics. Our choice of a uniform distribution for the variable labelled "age" was to demonstrate an effect of randomly generated numbers within a continuous variable, not to model the distribution of human age.

To ensure this is more clear to the reader, we have revised the manuscript. We have added the following lines to the Methods section [lines 582 - 585]:

“…we generated a set of arbitrary clinical phenotype variables. To minimise assumptions about the data's underlying distributions, each variable was modelled independently and drawn from uniform distributions.”

We have also clarified the arbitrary nature of the variable bounds by adding this sentence [line 594 - 595]:

“The lower and upper bounds for each variable were selected arbitrarily.”

Finally, we reinforce this point in the Results section with the following new text [lines 111 - 112]:

"For each patient, the clinical phenotype consists of multiple arbitrary clinical variables, including illustrative variables for..."

Regarding the number of variables, we intentionally restricted the number of variables to the minimum required to illustrate our core points. In drafting this manuscript, we found that models with more variables made the specific effects we were trying to highlight difficult for the reader to follow, without adding any new conceptual insights. The purpose was clarity of demonstration. To show that these principles are not limited to overly simplified scenarios, we included the analysis in Figure 6, which demonstrates that the same principles hold true in a model with much greater complexity, including a larger number of variables and covariance structures.

Regarding the use of a real-world database, the core purpose of this study was to demonstrate that certain machine learning techniques can detect treatment effects (and non-effects) that might be missed by traditional inferential statistics, we therefore chose simulated data to be able to program specific, known effects into the data and then definitively assess whether a given analytical technique can or cannot detect them. This provides a clear, unambiguous benchmark of the method's capabilities, which is not possible with a real-world dataset. Our aim here is to establish a principle of analytical sensitivity, which can then be applied to real-world data in future work.

We believe these revisions make the illustrative, non-realistic nature of our simulation explicit, directly addressing the reviewer's concerns and clarifying the methodological rationale for our readers.

An important criterion is missing: model power

We thank the reviewer for raising the important point about model power. This is an important criterion for establishing the reliability of the method, and we have now performed a new dedicated analysis to address it.

The analysis presented in Figure 3 was intended as a deep dive into the model's classification performance on a single, representative instance of our simulation. It demonstrates how the model works by more accurately identifying the ground truth than the noisy trial outcome but it does not establish the reliability of this success.

To address this, we have now conducted a formal power analysis to quantify the probability that our XGB approach will successfully detect the true treatment effect under the simulated conditions.

We defined "successful detection" in a given simulation run as the model achieving an accuracy of >90% against the known ground truth labels. We then ran our entire simulation and analysis pipeline 500 times, recording a success or failure for each run based on this criterion.

The results of this analysis show that the model's power was 100%. Specifically, in 500 out of 500 independent simulations, the XGB model achieved >90% accuracy against the ground truth, demonstrating that our method is highly reliable and not the result of a single, fortuitous outcome.

We have added details of this power analysis to the manuscript. In the Methods Machine learning analysis section [lines 676 - 679]:

“To quantify the statistical power and robustness of our analytical approach, the entire data generation and XGB analysis pipeline was repeated for 500 independent runs. For each run, a "successful detection" was defined as the model achieving an accuracy greater than 90% when classifying against the known ground truth.”

In the Results, Analysis of the same randomised control trial data using machine learning section [lines 237 - 239]:

“A power analysis revealed that the method was highly robust and a success criterion of >90% accuracy against ground truth was met in 100% of 500 simulation runs, resulting in a statistical power of 100% to detect the simulated effect under these conditions.”

For the LR model, why was the choice of L2 penalized? Since this is a simulation, an LR elastic model should have been used to determine the exact values of L1 and L2 that are preferable according to the simulated cohort.

Our choice of a standard L2-penalised model was a deliberate methodological decision, designed to create a robust and interpretable baseline for comparison against our primary XGB ML technique.

Our primary goal for the LR analysis was to create a strong, traditional statistical competitor. To this end, we did more than just fit a simple model; we explicitly enhanced its capability by including interaction terms between all combinations of variables. By giving the LR model the capacity to see these interactions, we sought to ensure a fair comparison with XGB.

With all variables and their interactions included, the primary risk to the model is overfitting due to multicollinearity and large coefficient values. L2 regularisation is precisely the standard and appropriate tool for managing this risk. It penalises large coefficients to prevent any single feature from dominating, which is exactly what is needed here. Therefore, a standard L2 penalty is the most parsimonious and directly applicable choice. We used the default, well-vetted parameters from the scikit-learn library. This approach represents a standard, reproducible, and widely understood implementation of logistic regression. Introducing a complex hyperparameter tuning process for the baseline model (as required by Elastic Net) would add a layer of optimisation that could obscure the core comparison of this study, which is focused on the fundamental differences between the analytical approaches, not on fine-tuning (and potential overfitting) a specific baseline model.

Our implemented LR model, enhanced with interaction terms and appropriately regularised with a standard L2 penalty, represents a powerful and fair baseline. It was intentionally configured this way to robustly test the hypothesis rather than being optimised for a single data instance. We believe this choice is methodologically sound and best serves the illustrative purpose of our study.

We have included mention of LR elastic models alongside discussion about other forms of hyperparameter tuning in the Discussion section [lines 513 - 517]:

“Applying these techniques effectively in practice requires systematic hyperparameter tuning (e.g., grid search, random search, Bayesian optimisation, LR elastic models) and sensitivity analysis to refine model performance. Although a detailed exploration is outside this work's scope, practical guidance on these techniques is readily available in other resources …”

Several studies have validated the power of XGBoost to predict the phenomena studied on real databases. However, an interpretability problem remains compared to its recognized flexibility. This problem is also present in the simulation carried out in this article. What is the added value of this work?

The reviewer is correct that while the predictive power of models like XGB is well-established, their perceived lack of interpretability has been a major barrier to clinical adoption. Addressing this very challenge is one of the central contributions of our work, i.e., providing a demonstration in how to deliver the clinical need for transparent, interpretable insights, rather than solely pursing predictive performance.

While many studies stop after reporting high accuracy, our paper's main purpose is to demonstrate a practical workflow for how this "black box" can be “opened” and its findings made useful to a clinician. We illustrate this through our application of post-hoc interpretability techniques, which allow us to move beyond a simple prediction and understand the drivers of that prediction.

We show how to determine the key factors driving the model's predictions across the entire cohort. For example, our analysis of feature importance in Figure 4 identifies which clinical phenotypes the model relied on most heavily, enabling clinicians to see whether biologically plausible factors are driving results and providing potentially hypothesis generating novel insights.

The added value is not necessarily in proving, once again, that XGB is powerful. The added value is in providing an illustrative template in how it can complement and directly address the flaws in standard inferential techniques used for clinical research. We demonstrate a practical methodology to translate a powerful but opaque prediction into a transparent, patient-specific insight that could guide clinical decision-making and hypothesis generation. Our work is intended to serve as a guide for researchers and clinicians on how to navigate the interpretability challenge, and to benefit from the full potential of these advanced models in medicine.

Our work also provides a practical demonstration of the data requirements needed for these methods to be reliable. We illustrate that the model's insights are highly sensitive to the completeness of the input data. The absence of even a single key feature can drastically alter the model’s performance, and the resulting clinical insights. This finding elevates the importance of comprehensive and robust patient phenotyping from merely "good practice" to an essential prerequisite for the safe and effective deployment of these advanced models.

It is important to note that the figures are of low resolution. Please increase it.

Thank you for noting the resolution issue. We have addressed this by providing a new, high-resolution version of Figure 4. All other figures have now been rendered at the maximum resolution and dimensions permitted by the journal's submission guidelines.

Reviewer #3: The methodology section used in this paper is missing. A well-developed and structured section is mandatory.

From the introduction section, we move on to the results section; without the methodology section, it is impossible to complete the article.

We direct the reviewer to the Methods section starting from page 14. The manuscript is structured in keeping with the journal’s manuscript organisation guidelines (https://journals.plos.org/plosone/s/submission-guidelines) whereby the Materials and Methods, Results and Discussion sections can be presented in any order. In the first paragraph of the Results section, we direct the reader to the Methods section for full methodological details.

Result section:

Title (lines 106-108): Too long as a title

We have revised this title to be more concise while retaining its core message. It now reads:

“Creation of simulated clinical cohort data and determinants of ‘ground truth’ treatment response”

Fig 1. Investigation outline line133 :

for figure 1: what is this explanatory and interpretative paragraph? make the interpretations of your text then put figure 1 in parentheses at the end of the paragraph

We have ensured that the Figure 1 legend adheres strictly to the journal’s figure caption requirements (https://journals.plos.org/plosone/s/figures#loc-captions), which mandate that captions must be self-sufficient and fully understandable without reference to the main text. To comply with the requirement to 'Describe each part of a multipart figure,' the current legend structure is necessary. Moving descriptive text elsewhere would compromise its standalone clarity and violate these guidelines.

title : An analysis of the randomised control trial data using traditional inferential statistics (lne 173 ) : Too long as a title

We have revised the section heading for brevity, ensuring it remains a clear guide to the content while maintaining consistency with other headings. It now reads:

“Analysis using traditional inferential statistics”

To maintain consistency, we also changed the title of the section which follows to:

“Analysis of the same data using machine learning”

Fig 2 (line 185-187) : is't title ?!

The same remarks for the rest of the figures and paragraphs

We have amended the caption for Figure 2 to clearly separate out a title and additional description in the legend. Upon reviewing all remaining figures, we confirmed that they were already consistent with this correct format, each featuring a distinct title. The issue was therefore seemingly isolated to Figure 2, which has now been rectified.

We were uncertain how the remark “is't (sic) title ?!” applies to 'paragraphs,' but if there is a specific formatting issue we have overlooked, we would be happy to address it.

Revise your text in the form of presenting the information so that it is easy for the reader to read

While we were encouraged that other reviewers complimented the writing, we recognise that clarity is paramount. In light of your feedback, we have carefully re-read the manuscript. It would be helpful to know if any lack of clarity stems from the writing itself, or perhaps from missing context which is detailed more fully in the Methods section which the reviewer’s first comment appears to indicate they had not been aware is included in the Manuscript. We would be grateful if you could point to any specific passages that are unclear, and we will happily revise them.

Reviewer #4: The revised manuscript "Machine learning detects hidden treatment response patterns only in the presence of comprehensive clinical phenotyping" is well written and an important contribution to the body of knowledge.

The authors responded well to all the criticisms and concerns raised by the reviewers and I feel the revised manuscript is fit for publication.

We thank the reviewer for their comments.

However, one minor observation is that the machine learning model as developed by the authors is generally more applicable to diseases without a definitive cause(s), for example degenera

---

## [Decision Letter · Decision Letter 2]

2 Oct 2025

Machine learning detects hidden treatment response patterns only in the presence of comprehensive clinical phenotyping

PONE-D-25-17285R2

Dear Dr. Auger,

We’re pleased to inform you that your manuscript has been judged scientifically suitable for publication and will be formally accepted for publication once it meets all outstanding technical requirements.

Kind regards,

Ziheng Wang

Academic Editor

PLOS ONE

Additional Editor Comments (optional):

Reviewers' comments:

Reviewer's Responses to Questions

**Comments to the Author**

Reviewer #2: All comments have been addressed

2. Is the manuscript technically sound, and do the data support the conclusions?

Reviewer #2: Yes

3. Has the statistical analysis been performed appropriately and rigorously?

Reviewer #2: Yes

4. Have the authors made all data underlying the findings in their manuscript fully available?

Reviewer #2: Yes

5. Is the manuscript presented in an intelligible fashion and written in standard English?

Reviewer #2: Yes

Reviewer #2: The authors have meticulously addressed all the concerns raised in my review. I recommend that this manuscript be accepted for publication in PLOS One.

We thank the reviewer again for their previous comments. We are pleased they are satisfied with the amendments.

**Do you want your identity to be public for this peer review?** For information about this choice, including consent withdrawal, please see our Privacy Policy

Reviewer #2: No

---

## [Editor Report · Acceptance letter]

PONE-D-25-17285R2

PLOS ONE

Dear Dr. Auger,

I'm pleased to inform you that your manuscript has been deemed suitable for publication in PLOS ONE. Congratulations! Your manuscript is now being handed over to our production team.

Kind regards,

on behalf of

Dr. Ziheng Wang

Academic Editor

PLOS ONE